

# A practical approach to lake water density from electrical conductivity and temperature

Santiago Moreira[1,2,3,*], Martin Schultze[1], Karsten Rahn[1] and Bertram Boehrer[1]

[1]UFZ-Helmholtz-Centre for Environmental Research, Department Lake Research, Brueckstrasse 3a, 39114 Magdeburg, Germany.

[2]Laboratoire des Sciences du Climat et de l'Environnement, LSCE/IPSL, CEA-CNRS-UVSQ, Université Paris-Saclay, F-91191 Gif-sur-Yvette, France

[3]Institute of Hydrobiology. TU-Dresden. Zellescher Weg 40, 01217 Dresden, Germany.

*Correspondence to*: S. Moreira (santiago.moreira-martinez@lsce.ipsl.fr)

**Abstract.** Density calculations are essential to study stratification, circulation patterns, internal wave formation and other aspects of hydrodynamics in lakes and reservoirs. Currently, the most common procedure is the use of CTD profilers and convert measurements of temperature and electrical conductivity into density. In limnic waters, such approaches are of limited accuracy, if they do not consider lake specific composition of solutes, as we show. A new approach is presented to correlate density and electrical conductivity, using only two specific coefficients based on the composition of solutes. First, it is necessary to evaluate the lake-specific coefficients connecting electrical conductivity with density. Once these coefficients have been obtained, density can easily be calculated based on CTD data. The new method has been tested against measured values and the most common equations used in the calculation of density in limnic and ocean conditions. The results show that our new approach can reproduce the density contribution of solutes with a relative accuracy of 10% in lake waters from very low to very high concentrations as well as in lakes of very particular water chemistry, which surmounts all commonly implemented density calculations in lakes by far. Finally, we provide a web link for downloading the corresponding density calculator.



# 1 Introduction

Density is one of main physical quantities governing the hydrodynamics, stratification, and mixing in lakes and reservoirs. Water quality in lakes is controlled by the biological and biogeochemical processes which depend on the availability of oxygen in the deep waters and nutrients in the surface waters. Both phenomena are controlled by the duration and extension

of the turnover period, which is dependent on density gradients. Therefore, density is a very important variable in numerical models for the simulation of the behaviour of lakes under changing conditions, e.g. due to management measures or phenomena related to Global Change.

The density of lake water (at atmospheric pressure) depends on temperature and dissolved water constituents. Since

temperature, composition and concentrations may vary over time, from lake to lake or even within a particular lake due to seasonal stratification or meromixis, numerical models of lakes calculate the density internally. There are several approaches to calculate water density in lakes. Most of them are general equations that not always reflect specific properties of lakes. If enough measurements of density for the relevant temperature range are available and composition and concentrations of the main constituents are constant, regressions can be used to generate a mathematical formula for density in a specific lake (e.g.

Jellison et al., 1999; Vollmer et al., 2002; Karakas et al., 2003). If the composition is constant and the main constituents are ions, electrical conductivity or salinity may be used as an easy to measure proxy for concentrations (Bührer and Ambühl, 1975; Chen and Millero, 1986; Pawlowicz and Feistel, 2012).

Imboden and Wüest (1996) discussed the influence of dissolved substances on (potential) density because not only the

concentration of the total dissolved solids changes considerably from lake to lake but also the chemical composition (see e.g. Boehrer and Schultze, 2008). The effects of dissolved solids on density stratification have been studied in lake specific investigation in Lake Malawi (Wüest et al., 1996) and in Lake Matano (Katsev et al., 2010). In some cases, the specific contribution of ions such as calcium, carbonate or dissolved iron can control the permanent stratification in lakes as in La Cruz (Spain) (Rodrigo et al., 2001), Cueva de la Mora (Spain) (Sanchez-España et al., 2009) or Waldsee (Germany) (Dietz et

al., 2012).

Density of pure water can be calculated using mathematical expressions such as Kell (1975) or Tanaka et al. (2001). Density calculations of natural waters require additional terms to include the contributions of dissolved substances. Specific formulas have been developed for ocean conditions. The UNESCO equations developed by Fofonoff and Millard (1983) have been

the standard for a long period. They used temperature and practical salinity based on electrical conductivity measurements. Because sea water conditions are a known reference and the approaches provide stable results over a wide range of temperatures and electrical conductivity, these have been applied in limnic systems and implemented in numerical models such as DYRESM (Imberger and Patterson, 1981; Gal et al., 2009; Imerito, 2014), ELCOM (Hodges and Dallimore, 2007),





GOTM (Burchard et al., 1999; Umlauf et al., 2005) or CE-QUAL-W2 (Cole and Buchak, 1995). Recently the ocean standard was replaced by the new Thermodynamic Equation of Seawater 2010 (TEOS-10; IOC et al., 2010). Pawlowicz and Feistel (2012) have considered the application of TEOS-10 (IOC et al., 2010) in several cases different from seawater.

Highly accurate electrical conductivity measurements and temperature provide an easy to implement approach to density. Consequently, such formulas have been used for limnic waters as well. As the composition of solutes differs greatly from the ocean, density calculation based on ocean conditions can only be of limited accuracy. Bührer and Ambühl (1975) developed an equation to calculate density based on temperature and specific conductance at 20 ºC for alpine lakes. In addition, a popular approach was formulated by Chen and Millero (1986) tuning ocean approaches to freshwater conditions (salinity <

0.6 psu).

Higher accuracy can be achieved when site specific density equations are produced. Jellison et al. (1999) developed a density equation for Lake Mono from water samples which have been measured at different temperatures and dilutions. In the case of meromictic lakes, strong differences in the composition of the mixolimnion and monimolimnion must be reflected in the

density equations in order to sustain the permanent stratification in the density calculations. Boehrer et al. (2009) and von Rohden et al. (2010) used an equation based on density measurements of the monimolimnion and mixolimnion of Lake Waldsee.

Boehrer et al. (2010) evaluated the contribution of the different cations and anions separately in terms of the partial molal

volumes and implemented an algorithm, RHOMV (http://www.ufz.de/webax), to calculate density with a second order approximation for temperature dependence and ionic strength dependence.  Pawlowicz et al. (2011) implemented the LIMBETA method that calculates density from composition. Another approach comes from Pawlowicz et al. (2012) where the authors propose to use TEOS-10 but replace seawater salinity by specific salinities obtained and corrected for freshwaters. This limnic salinity can be calculated using the chemical composition by summing up all the dissolved solutes

($Sa^{soln}$) or by summing up only the dissolved ions ($Sa^{ionic}$) and correcting this value with the dissolved $Si(OH)_4$, $Sa^{dens} = Sa^{ionic}$ $+ 50.6 \times [Si(OH)^{-4}]$ (mol kg$^{-1}$).

Based on partial molal volumes (RHOMV), Dietz et al. (2012) separated the contributions of solutes for freshwater lakes. Moreira et al. (2011) based density on the composition of solutes in their model to reproduce the permanent stratification of

Lake Waldsee numerically using RHOMV to include the reactivity of substances in the density (see also Nixdorf and Boehrer 2015). In conclusion, we see the necessity of including the chemical composition to provide a reasonably accurate density formula. However, we accept the need for a practical density approach, which can easily be implemented, such a mathematical formula that relates density to temperature and electrical conductivity.



In this publication, we propose to develop coefficients for such a formula from the chemical composition. We provide an algorithm RHO_LAMBDA (from $\rho_\lambda$) to obtain such coefficients and demonstrate the applicability of the approach with water from Rappbode Reservoir. We also deliver an appropriate assessment for the Rappbode Reservoir case and compare the accuracy with other approaches currently in use for limnic waters. For a quantitative judgement of the applicability of our

5    approach, we also evaluate coefficients for two further fresh water bodies (Lake Geneva, Lake Constance), an extremely saline lake (Lake Mono), a meromictic open pit lake in the mixolimnion and the monimolimnion (Lake Waldsee) and finally sea water as a globally known example and well defined standard.

**2 Methods: The proposed approach (RHO_LAMBDA)**

We propose a simple equation for density as a numerical approximation of the (potential) density of lake water:

$$\rho \approx \rho_\lambda\left(T, \kappa_{25}\right) = \rho_w(T) + \kappa_{25}\left[\lambda_0 + \lambda_1 \times \left(T - 25\,°C\right)\right] \qquad (1)$$

where the first term of the right side $\rho_w$ corresponds to density of pure water, which can be calculated in a very accurate way using Kell (1975) or Tanaka (2001). Our approach (Eq. (1)) correlates density to temperature (T) and electrical conductivity at 25°C ($\kappa_{25}$) of a water sample using coefficients $\lambda_0$ and $\lambda_1$. The introduction of $\lambda_1$ can reflect temperature dependence, which is required for a shifting temperature of maximum density. Only two coefficients need to be determined, and thus this

equation is easy to implement.  Coefficients $\lambda_0$ and $\lambda_1$ can be obtained as follows.

At $T$=25°C, the $\lambda_1$ term in Eq. (1) vanishes and $\lambda_0$ can be calculated using Eq. (2) provided that the water density at 25°C is known from other sources:

$$\lambda_0 = \frac{\rho\left(T = 25\,°C, \kappa_{25}\right) - \rho_w\left(T = 25\,°C\right)}{\kappa_{25}} \qquad (2)$$

If density is known for a temperature $T \neq 25°C$, $\lambda_1$ can be calculated in a second step:

$$\lambda_1 = \frac{\left[\rho\left(T, \kappa_{25}\right) - \rho_w(T)\right]/\kappa_{25} - \lambda_0}{T - 25\,°C} \qquad (3)$$

Necessary data for equations 2 and 3 can be derived from measurements or from calculations.

In our RHO_LAMBDA approach, we use Tanaka (2001) equation for pure water density, $\rho_w$. If the composition of solutes in

the water is known, density of water is calculated by using RHOMV (Boehrer et al., 2010) and finally $\kappa_{25}$ is provided by the





algorithm implemented in the PHREEQC code (Parkhust and Appelo, 1999) and whose description can be found in Atkins and de Paula (2009). The method is also described in detail in Appelo's webpage of PHREEQC, http://www.hydrochemistry.eu/exmpls/sc.html. This method (re-implemented in Python from the original code) calculates the specific conductance of a solution from the concentration, the activity coefficient and the diffusion coefficient of all the

charged species. The diffusion coefficients can be found in Millero (2001).

## 2.1 Rappbode Reservoir

We demonstrate our density approach with the example of Rappbode Reservoir (Germany; for details on this reservoir see Rinke et al. 2013 and references therein): its low electrical conductivity indicates low concentrations of solutes. From

chemical analysis of a surface sample from 19 November 2010, we knew major cations were calcium (13.8 mg L$^{-1}$) and sodium (9.3 mg L$^{-1}$), while major anions were bicarbonate (28.07 mg L$^{-1}$), sulphate (18.5 mg L$^{-1}$) and chloride (16.8 mg L$^{-1}$) (see Table 1). In addition, a considerable portion of organic matter (3.1 mg DOC L$^{-1}$) and silicate (4.5 mg L$^{-1}$ of $Si(OH)_4$) were contained in the sample.

1)    For this sample, an electrical conductance $\kappa_{25}$ = 0.1634 mS cm$^{-1}$ was calculated by inserting given concentrations

into the PHREEQC algorithm (Parkhust and Appelo, 1999; Atkins and de Paula, 2009).

2)    According to RHOMV, the density of this sample at 25 ºC was $\rho_{MV}(T=25°C)$ = 997.130 kg m$^{-3}$ and $\rho_w(T=25°C)$ = 997.047 kg m$^{-3}$.

3)     Putting these numbers into Eq. (2) delivered $\lambda_0$ = 0.506 kg cm m$^{-3}$ mS$^{-1}$

4)    Similarly we evaluated $\lambda_1$ = -0.00115 kg cm m$^{-3}$ mS$^{-1}$ K$^{-1}$ by putting $\rho_w(T=5°C)$ = 999.967 kg m$^{-3}$ and $\rho_{MV}$

$(T=5°C)$ = 1000.053 kg m$^{-3}$ into Eq. (3).

5)    Finally inserting the lambdas as coefficients into Eq. (1) delivered a density formula for Rappbode Reservoir.



## 3 Assesments

The practicability of this approach depends on its accuracy. This will first be assessed for Rappbode Reservoir water and its above evaluated coefficients. However, for limnologists working on other limnic water bodies, an assessment of accuracy in the general range of limnic water composition is of fundamental interest. In conclusion, we chose a collection of lake waters of different chemical composition and a wide range of concentrations. We included all lakes we knew of, where a reliable reference density could be provided, and the chemical composition was known.

In specific, we included two further typical freshwater lakes, Lake Geneva and Lake Constance, which are well known in the limnological literature. As an example for saline lakes, we chose Lake Mono (eg. Jellison et al., 1999). In order to include also water with rather unusual composition, we chose two water samples from a meromictic open pit lake, Lake Waldsee, which contains large amounts of sulphate and dissolved iron (e.g. Dietz et al., 2008, 2012; Boehrer et al., 2009, von Rohden et al., 2010; Moreira et al., 2011). Finally, we used seawater as a reference water, of which the composition is known at high accuracy. Table 1 presents the original chemical compositions of the different lakes considered in the testing of the RHO_LAMBDA expression. Data were derived from chemical analysis (for experimental details see Appendix) or literature (for references see Table 1). We complemented the set by synthetically produced lake water of differing composition and concentration from the work by Gomell and Boehrer (2015) in order to test systematically the influence of composition and concentration on the values of the coefficients $\lambda_0$ and $\lambda_1$ (for experimental details see Appendix).

For critical comparison with other density equations, our assessment section (Sect. 3) consists of two major parts: first we check the accuracy for different lakes and water samples and secondly we provide the lambda coefficients of several aquatic systems where we have direct measurements or an alternative approach to density to check the accuracy of $\rho_\lambda$ in general. Table 2 presents the results of the intermediate step calculations to obtain $\lambda_0$ and $\lambda_1$. As references for the assessment, we used the measured (for details see Appendix) or published data (Figure1, Table 2).

The quantitative comparison between the different methods (including RHO_LAMBDA) and the reference values is shown in Fig. 1. Our approach mainly aimed at representing the density contribution of solutes. Hence we related the difference to our reference with the contribution of the solutes

$$\text{Rel. Error} = \left(\rho_\lambda - \rho_{\text{ref}}\right) / \left(\rho_{\text{ref}} - \rho_w\right) \qquad (4)$$



For any temperature within the range of 1-30°C defined according to typical limnic conditions, the values of the relative error defined by Eq. (4) are displayed in the right column of Fig. 1. On purpose, we obtained the chemical composition from a different source (sample) than the density measurement. Hence the error of variable water composition within one lake was
included in our assessment.

To judge the accuracy of our approach, we also inserted results from other formulas in common use for transferring CTD data into density: we included UNESCO (Fofonoff and Millard, 1983), TEOS-10 (IOC et al., 2010), Chen and Millero (1986) and Bührer and Ambühl (1975) (Fig. 1) as far as possible according to the defined range of applicability of the single
formula.

**Rappbode Reservoir:** The measured conductance ($\kappa_{25}$) of our Rappbode Reservoir sample was 0.1579 mS cm$^{-1}$, which differed only by 4% from the value 0.1635 mS cm$^{-1}$ calculated using the PHREEQC electrical conductivity algorithm at 25°C (described in Atkins and De Paula 2009). This is within the measurement accuracy of the chemical analysis. Reference
density was produced by measuring in a densitometer PAAR DSA 5000 from 1 to 30°C.

We can see that the RHO_LAMBDA method reproduced the reference values of the water sample from Rappbode Reservoir with a relative error ranging from -12.7% to 4.3%. The deviation from the reference was even lower than 5% in the range 10 to 27 ºC. Among the other compared approaches, TEOS-10 showed the best results with relative error ranging from -15.7% to 0.1%. The Bührer and Ambühl (1975) approach resulted in relative error ranging from -4.0% to 99.3% and strongly rising
with temperatures increasing above 20°C. Results according to Chen and Millero (1986) ranged between -37.8% and -25.3%.

**Lake Geneva:** Calculated and measured electrical conductivity ($\kappa_{25}$) of a water sample from 07.11.2013 differed by less than 1% for 25°C (Table 2). Reference density was produced from this sample in a PAAR DSA 5000 densitometer. The relative
error ranged from -11.5% to -4.6% for our RHO_LAMBDA approach. Bührer and Ambühl (1975) (relative error -15.3% to 21.7%), TEOS-10 (relative error -12.9% to -7.8%) and Chen and Millero (1986) (relative error -50.9% to -47.6%) showed larger deviations from the reference.

**Lake Constance:** The composition shown in Table 1 mainly coincided with the analysis done by Stabel (1998). The
calculated conductivity at 25 ºC ($\kappa_{25}$) of 0.330 mS cm$^{-1}$ differed from the measured value of 0.322 mS/cm around 3 %. Also





here reference density was from measurements in a PAAR DSA 5000 densitometer. The relative error ranged from -9.7% to -4.2% for RHO_LAMBDA approach. TEOS-10 (relative error -12.2% to -8.6%), Bührer and Ambühl (1975) (relative error -17.4% to 14.6%) and Chen and Millero (1986) (relative error -46.6% to -44.1%) had again larger deviations from the reference. The strong increase of the relative error of Bührer and Anbühl (1975) with temperature was smallest for Lake

5 Constance compared to the other freshwater lakes.

**Lake Mono:** We evaluated density for a water sample of conductivity of $\kappa_{25}$ = 85.67 mS cm$^{-1}$ which was provided by Jellison et al. (1999) and differed 12 % from the calculated value 96.61 mS/cm using the PHREEQC algorithm. The density formula by Jellison et al. (1999) was used as the reference density. In this case, the relative error using RHO-LAMBDA ranged from

10 -9.5% to -1.5% even in this lake with such saline waters and unusual composition. Also in this case, TEOS-10 showed larger deviation from the reference (relative error -10.4% to -5.2%). The largest relative error was found for the UNESCO equation according to Foffonof and Millard (1983) (relative error -39.9% to -36.0%).

**Lake Waldsee mixolimnion / Lake Waldsee monimolimnion:** This case presented a meromictic open pit lake (Boehrer et

15 al., 2008; Dietz et al., 2008, 2012; von Rohden et al., 2010; Moreira et al., 2011) of moderate salinity (0.22 psu in the mixolimnion and 0.6 psu in the monimolimnion, Moreira et al. 2011), but its composition differed from the usual carbonate or chloride waters. Composition was obtained from Dietz et al. (2008, 2012). A correction for DOC content was also introduced according to Dietz et al. (2012). This correction increased density by 0.015 kg m$^{-3}$ in the mixolimnion and by 0.06 kg m$^{-3}$ in the monimolimnion.

The calculated $\kappa_{25}$ differed 7.0% from the reference value in the mixolimnion and 7.6% in the monimolimnion (Table 2). This was the highest difference between reference and calculated value of all waters considered in this study. Probably, the very special chemical composition of the waters was the reason. The missing data for ammonia and silicate may also have contributed, in particular in the monimolimnion. Measurements in the work Boehrer et al. (2009) were used as density

25 reference.

The relative error of the RHO-LAMBDA approach ranged from -8.4% to -3.9% in the mixolimnion. In the monimolimnion, the relative error ranged from -11.9% to -9.8% for the RHO_LAMBDA approach. The deviation from the reference was substantially larger for all other compared approaches (Figure 1). The averages of the absolute values of the relative error

30 were 22.8%, 52.0% and 52.3% for TEOS-10 (IOC et al., 2010), Chen and Millero (1986) and UNESCO (Foffonof and



Millard, 1983) in the mixolimnion, respectively. In the monimolimnion, the values were 35.2% for TEOS-10, 60.0% for Chen and Millero (1986) and 60.2% for UNESCO (Foffonof and Millard, 1983).

**Seawater:** The seawater composition was obtained from Millero et al. (2008) and we used TEOS-10 (IOC et al., 2010) as our sea water density reference. Electrical conductivity was calculated for this composition and resulted in 53.76 mS cm$^{-1}$, while the reference value given by Millero et al. (2008) was 53.06 mS cm$^{-1}$. That meant the deviation was 1.3%. As expected – both formulas were specifically designed for ocean water – , the relative error of the UNESCO approach according to Foffonof and Millard (1983) was very small, ranging between -0.02% and -0.01%. This was probably a result of numerical uncertainties of the calculations. The relative error of our RHO-Lambda approach ranged between -0.75% and 0.68%.

## 4 Discussion

In all cases, our density approach reproduced the density contribution of the salts within 10%. This is better than most of other of the approaches, which differed up to 60% from the correct values. Even in the case of very low concentrations (Rappbode Reservoir) and very high concentrations (Lake Mono) as well as in very special water composition (mine lake Waldsee), the 10% accuracy for the salt contribution was achieved with our RHO_LAMBDA approach. The observed strong increase of the relative error with temperature for Bührer and Ambühl (1975) was caused by the inclusion of temperatures above 24°C. Bührer and Ambühl (1975) used an equation for the density of pure water which was applicable from 0°C to 24°C. As a consequence, the relative error for Bührer and Ambühl (1975) becomes smaller with increasing overall content of dissolved ions.

The first coefficient $\lambda_0$ varied over the range from 0.37 to 0.88 kg cm m$^{-3}$ mS$^{-1}$, see Fig.2. This was more than a factor of 2 and explains that a density formula with constant coefficients could never be able to mimic density accurately for a larger range of lake waters. Obviously the coefficient $\lambda_0$ depended on the composition of the solutes. A dominance of double charged ions – opposed to single charged ions – leads to higher values of $\lambda_0$. This effect was clearly visible in the inclusion of calculated values for a NaCl solution and a CaSO$_4$ solution of 1g L$^{-1}$ each in the display (Fig. 2). $\lambda_0$ differed nearly by a factor of 2.

Also the concentration of solutes had a decisive effect on the coefficients. We used density measurements of a dilution series of synthetic lake waters by Gomell and Boehrer, (2015) of 1, 3, 10, 30, or 90 g L$^{-1}$ of a mixture of KCl, NaHCO$_3$ and Na$_2$SO$_4$. We included lambda coefficients from the RHO_LAMBDA approach "Mix" together with regressions of the





published measured data "Mix-M" (Fig. 2). Both empirical data as well RHO_LAMBDA results reflected the concentration effect on $\lambda_0$ of a factor of 1.5. Although not perfect, the agreement between empirical data and RHO_LAMBDA values lay within the 10% margin we found for lake waters above.

Values for $\lambda_1$ nearly all lay between -0.001 and -0.002 kg cm m$^{-3}$ mS$^{-1}$ K$^{-1}$). Hence the $\lambda_1$ term delivered a small contribution in all cases, i.e. always an order of magnitude smaller compared to the $\lambda_0$ term. As a consequence, it could be neglected for most limnological applications. Though not really necessary for an absolute density calculation, $\lambda_1$ was included to also represent the shift of temperature of maximum density for a given lake water composition, which could not be achieved with the $\lambda_0$ term alone. Negative values of $\lambda_1$ indicated a shift of the temperature of maximum density to lower temperatures. A

closer look at the $\lambda_1$ values revealed that some empirical values (also Lake Mono reference derived from empirical measurements) lay considerably lower than expected from coefficients of physical chemistry. Large differences were only concerned with freshwater lakes, hence the temperature of maximum density was not really affected in those cases. However, the difference posed the question how accurately the shift of temperature of maximum density would actually be indicated by coefficients of physical chemistry literature.

**5 Comments and Recommendations**

We showed that the correlation between electrical conductivity and density depends strongly on composition and concentration of solutes. As a consequence, the limnic range cannot be covered with one formula with constant coefficients. However, a simple mathematical addition of two terms to a pure water formula is able to represent the density contribution of solutes in all our examples at an accuracy of 10%. This is sufficient for most limnological applications and is better than

any other density approach based on CTD data, if not specifically designed for a given lake water.

Only two coefficients $\lambda_0$ and $\lambda_1$ need to be evaluated: While $\lambda_0$ varies considerably between lakes, the numerical evaluation of $\lambda_1$ delivers very similar values of $\lambda_1 \sim -0.0015$ kg cm m$^{-3}$ mS$^{-1}$ K$^{-1}$ for any lake water composition. Hence once $\lambda_0$ has been evaluated for a lake, a rather accurate and simple density formula can be used for CTD data. The approach uses conductance

$\kappa_{25}$, which can be verified in limnic waters, and thus avoids salinity, which is a delicate quantity in limnic waters. The inclusion of such a simple and rather accurate approach for potential density in numerical lake models would be desirable.

For the convenient use and implementation, a density calculator tool is provided on
https://sourceforge.net/projects/densitycalc





**Appendix *A***

**Measurement of samples taken for this study**

All samples were taken as surface samples and stored cooled and without bubbles in polyethylene bottles until measurements and analysis in the lab.

Density measurements were done in 1°C steps between 1°C and 30°C using a PAAR DSA500 densitometer. Measurements of electrical conductivity were done with a MultiLab-Pilot conductivity meter (WTW, Germany).

pH was measured using a HQ11d pH-meter (Hach-Lange, Germany) in the lab. Sulphate ($SO_4^{2-}$), and chloride ($Cl^-$) were

10    analyzed by suppressed conductivity using an ICS-3000 ion chromatography system (Dionex, Idstein, Germany) and automatically generated potassium hydroxide eluent. Concentrations of Ca, Mg, Na, K, Al, Fe, and Mn were determined by optical emission spectroscopy with inductively coupled plasma (ICP-OES, Perkin-Elmer, OPTIMA 3000, Germany) (Baborowski, et al., 2011). Acidity and alkalinity were measured by an automatic titrator (Metrohm, Germany). Bicarbonate and carbonate were calculated based on acidity, alkalinity and pH using PHREEQC (Parkhust and Appelo 1999).

Nitrate ($NO_3^-$) (DIN_EN_ISO_13395, 1996; Herzsprung, et al., 2005), ammonium ($NH_4^+$) (Krom, 1980; DIN_EN_ISO_11732, 1997), and silicate ($Si(OH)_4$) (Smith & Milne, 1981) were measured by continuous flow analysis (CFA, Skalar, The Netherlands) (Herzsprung, et al., 2006).

20    Fluoride ($F^-$) and borate ($B(OH)_4^-$) were not included into the analyses because they usually are not relevant for density in typical freshwater lakes.

**Corrections of original chemical analyses for charge balance**

If the charge balance between cations and anions was higher than 5% or below -5%, the concentrations of cations were

25    increased or diminished to reach balance by keeping the ratios of the cations to each other constant. The following corrections were necessary: reduction by 16% for Lake Geneva, reduction by 10.4% for Lake Constance, increase by 15% for Lake Mono, increase by 7% for mixolimnion and reduction by 15% for monimolimnion of Lake Waldsee.





### Application of TEOS10 algorithm in the Assessment

The initial algorithm of TEOS10 according to IOC et al. (2010) was applied only for seawater serving as reference. In all other cases, the adaptation for limnic systems proposed by Pawlowicz and Feistel (2012) was used since all other systems are

limnic. Because the only difference between both algorithms is the calculation of the so-called absolute salinity and the equation for density is the same, "TEOS-" was used in the legends of all diagrams in Figure 1.

### Preparation of synthetic solutions

For systematic investigation of dependencies of coefficients $\lambda_0$ and $\lambda_1$ we prepared solutions of pure NaCl (1 g L$^{-1}$) and pure

CaSO$_4$ (1 g L$^{-1}$) and proportional mixtures of KCl, NaHCO$_3$ and Na$_2$SO$_4$ having overall concentrations of 1, 3 ,10, 30 and 90 g L$^{-1}$. The water samples are labelled using the chemical formula of the salts (NaCl; CaSO$_4$) and as "MixN" with N being a number indicating the concentration. More detail about these prepared solutions can be found in Gomell and Boehrer (2015).

### Software

All the density methods have been implemented in Python 2.7 except the TEOS-10 (IOC et al., 2010). For TEOS-10 the original Fortran 90 library has been downloaded from http://www.teos-10.org/software.htm and compiled using f2py. The generated Python library has been used directly for the calculations using the Python 2.7 scripts. All the results presented in this manuscript can be obtained using the "density calculator" provided in https://sourceforge.net/projects/densitycalc

### Acknowledgements

We thank Ulrich Lemmin for taking and sending a water sample from Lake Geneva, and Karsten Rinke for a water sample from Lake Constance.



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





Table 1. Original chemical composition of the water in different test cases presented in the Sect. 3. All values except pH are presented in mg L$^{-1}$. (NA – not analysed)

| | Reservoir Rappbode | Lake Geneva | Lake Constance | Lake Mono | Waldsee (mixo.) | Waldsee (monimo.) | Seawater |
|---|---|---|---|---|---|---|---|
| pH | 7.14 | 7.0 | 7.9 | 9.8. | 7.1 | 6.7 | 7.0 |
| Na$^+$ | 9.30 | 11.1 | 5.60 | 32933.18 | 9.66 | 10.81 | 10919.56 |
| K$^+$ | 1.00 | 1.74 | 1.48 | 1610.92 | 7.04 | 10.17 | 404.23 |
| Ca$^{2+}$ | 13.8 | 44.30 | 51.2 | 6.01 | 61.32 | 89.78 | 417.38 |
| Mg$^{2+}$ | 3.30 | 6.49 | 9.03 | 31.59 | 12.88 | 17.50 | 1299.88 |
| NH$_4^+$ | 0.03 | <0.010 | <0.010 | NA | NA | NA | NA |
| Fe | NA | <0.025 | <0.01 | NA | 0.50 | 131.79 | NA |
| Fe$^{2+}$ | 0.00 | NA | NA | NA | 0.22 | 117.83 | NA |
| Fe$^{3+}$ | NA | NA | NA | NA | 0.28 | 13.96 | NA |
| Mn$^{2+}$ | 0.004 | <0.010 | <0.007 | NA | 0.22 | 0.88 | NA |
| Al$^{3+}$ | 0.01 | <0.005 | <0.02 | NA | NA | NA | NA |
| F$^-$ | 0.00 | NA | NA | NA | NA | NA | 1.31 |
| Cl$^-$ | 16.8 | 10.39 | 7.81 | 19043.74 | 5.67 | 4.96 | 19598.77 |
| SO$_4^{2-}$ | 18.5 | 44.56 | 33.26 | 10912.64 | 184.44 | 176.75 | 2747.05 |
| NO$_3^-$ | 6.50 | 0.483 | 3.37 | 0.00 | 2.85 | 1.24 | NA |
| HCO$_3^-$ | 28.07 | 94.58 | 136,68 | 3276.67 | 57.97 | 374.04 | 106.15 |
| CO$_3^{2-}$ | NA | 0.000 | NA | 17726.95 | 0.00 | 0.00 | 14.53 |
| Si(OH)$_4$ | 4.50 | 0.582 | 4.42 | NA | NA | NA | NA |
| B(OH)$_4^-$ | NA | NA | NA | 752.45 | NA | NA | 4.23 |
| Correction* | 0% | -16% | 10.4% | 15% | 7% | -15% | 0% |
| Source of data | Measured | Measured | Measured | Jellison et al. (1999) | Dietz et al. (2008) Dietz et al. (2012). | | Millero et al. (2008) |

*  *Correction of cation concentrations for charge balance*



**Table 2. Summary of the calculated values for obtaining RHOMV_LAMBDA coefficients. Lambdas and references. $\lambda_0$ and $\lambda_1$ represent the lambda values obtained using the chemical composition of Table 1 to calculate density and conductivity at 5 and 25 ºC, while $\lambda_0^*$ and $\lambda_1^*$ represent the lambda values obtained from a linear regression of the density reference. $\lambda_0$ and $\lambda_0^*$ are expressed in kg cm m$^{-3}$ mS$^{-1}$ and $\lambda_1$ and $\lambda_1^*$ are expressed in kg cm m$^{-3}$ mS$^{-1}$ K$^{-1}$. Density values are expressed in kg/m$^3$. Corrected Salinity shows the values of Practical salinity correct by a factor of 1.00488 for Chen and Millero (1986) method. For more details see text.**

| | Reservoir Rappbode | Lake Geneva | Lake Constance | Lake Mono | Waldsee (mixo.) | Waldsee (monimo.) | Seawater |
|---|---|---|---|---|---|---|---|
| $\kappa_{20}/\kappa_{25}$ (measured) ($\mu$s cm$^{-1}$) | 142.0/ 157.9 | 263/ 294 | 302/ 333.7 | --/ 85668 | --/ 550 | --/ 1050 | --/ 53064.9 |
| $\kappa_{25}$ (calc.) ($\mu$s cm$^{-1}$) | 163.49 | 296.81 | 329.768 | 96609.5 | 588.50 | 969.926 | 53762.53 |
| Practical Salinity | 0.0770 | 0.127 | 0.1613 | 83.04 | 0.220 | 0.60 | 35 |
| Absolute Salinity | 0.0997 | 0.22 | 0.251 | 91.11 | 0.351 | 0.78 | 35.165 |
| Corrected Salinity | 0.0774 | 0.128 | 0.1621 | 83.45 | 0.221 | 0.6029 | 35.171 |
| $\rho_{MV}$ (T=25°C) | 997.130 | 997.222 | 997.252 | 1075.480 | 997.383 | 997.744 | 1023.662 |
| $\rho_{MV}$ (T=5°C) | 1000.053 | 1000.149 | 1000.181 | 1083.661 | 1000.316 | 1000.691 | 1028.150 |
| $\rho_{ref}$ (T=25°C) | 997.126 | 997.228 | 997.253 | 1069.936 | 997.391 | 997.958 | 1023.344 |
| $\rho_{ref}$ (T=5°C) | 1000.059 | 1000.168 | 1000,194 | 1075.447 | 1000.332 | 1000.923 | 1027.6 |
| Data Sources | Measured | Measured | Measured | Jellison et al. (1999) | Dietz et al. (2008). Dietz et al. (2012). von Rohden et al. (2010) Moreira et al. (2011) | | Millero et al. (2008) |
| $\lambda_0$ | 0.50587 | 0.58947 | 0.62134 | 0.81185 | 0.59636 | 0.78061 | 0.49504 |
| $\lambda_1$ | -0.00115 | -0.00129 | -0.00135 | -0.00272 | -0.0011 | -0.0014 | -0.00146 |
| $\lambda_0^*$ | 0.5003 | 0.616 | 0.640 | 0.8508 | 0.6254 | 0.8676 | 0.495 |
| $\lambda_1^*$ | -0.00418 | -0.0034 | -0.0033 | -0.00151 | -0.00193 | -0.00215 | -0.0013 |
| A.E.[1] | 4.3% | 6.9% | 5.9% | 5.5% | 5.2% | 10.4% | 0.4% |
| M.E.[2] | 12.7% | 11.5% | 9.7% | 9.5% | 8.4% | 11.85% | 0.75% |

[1] Average absolute value of relative error. [2] Max. Absolute value of relative error.





Figure 1: Test cases (part I). Left panel presents the density curves of the different methods and the right panel shows the relative error of the density contribution of the solutes respect to the reference. In all the cases, measured values have been used as reference except in the cases of the Lake Mono (Jellison et al., 1999) and seawater (TEOS-10, IOC et al., 2010) which use specific density equations.





Figure 1b: Test cases (part II)





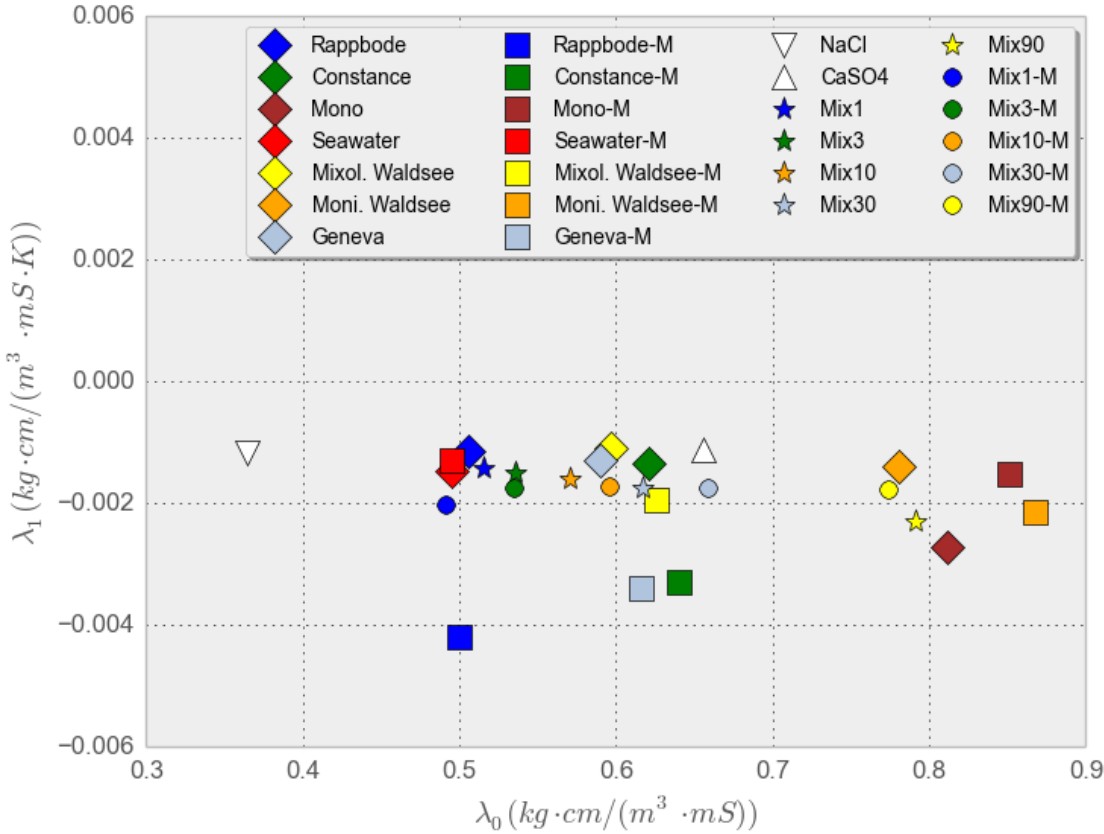

Figure 2. Distribution of the values of $\lambda_1$ versus $\lambda_0$. Concentrations for NaCl and CaSO$_4$ are 1 g/L in both cases. Chemical composition for the lakes and seawater are presented in Table 1. The water samples labelled as "Mix" are proportional mixtures of KCl, NaHCO$_3$ and Na$_2$SO$_4$ of 1, 3 ,10, 30 and 90 g/L (Gomell and Boehrer, 2015) and the water samples labelled as "-M" correspond to the lambda coefficients obtained from direct measurements of density and conductivity.