# Peer review of "A practical approach to lake water density from electrical conductivity and temperature"

_Hydrology and Earth System Sciences, 2016_

## Short Comment (SC1) · 15 Mar 2016

**HESS Title: A practical approach to lake water density from electrical conductivity and temperature**
**Author(s): Santiago Moreira et al. MS No.: hess-2016-36**
**MS Type: Research article**

As a physical limnologist researching on deep lakes, I know the importance of correctly evaluating the equation of state for the waters of the lake. For this reason I have reviewed with great interest the paper by Moreira et al. and my final evaluation is positive.
The topic is fundamental, the paper is clear and reasonably well written. The paper is neat, with no useless redundancy.
Probably the title and the abstract create too many expectations: one might expect that, after calibration for a lake, simply measuring conductivity and temperature profiles could provide density profile. As far as I understand, this is true only if the chemical composition does not change over the water column. Actually, eq. (2) depends on the composition of solutes so that, if this quantity keeps changing over the depth, as it happens in weakly mixed deep lakes, it should be recomputed for each depth on the basis, e.g., of RHOMV. Can the Authors better explain this point in the Discussion ?

I suggest accepting the paper after improving some editing problems that spoil its quality and some typos here and there: both of them I list in the folowing.

List of Typos:

| pag | line | part to be changed | change as |
|-----|------|--------------------|-----------|
| 2 | 33 | ELCOM (Hodges and Dallimore, 2007) | in reality in the References is 2014 |
| 3 | 26 | $Si(OH)^{-4}$ | $Si(OH)_4$ |
| 4 | 24 | Add, for clarity's sake: " in the remaining part of this paper, eq(1), completed by eq(2) and eq(3), will be referenced a RHO_LAMBDA approach." | |
| 5 | 14 | . $0.1634\,mS\,cm^{-1}$ | According o the data of Table 2, the last significant digit should be 5 |
| 5 | 19 | . Here Is $\varrho_{MV}$ computed according to RHOMV ? | |
| 6 | 19 | Probably in place of "our assessment section (Sect. 3)" "this section" would be better. | |
| 6 | 21 | This phrase could be improved, such as "an alternative approach to compute density, in order | |

| | | | |
|---|---|---|---|
| | | to check the accuracy of $\varrho_\lambda$." | |
| 7 | 3-5 | "Hence...". Could you kindly explain better ? | |
| 9 | 11 | most of other of the approaches | most of the other approaches |
| 15 | 24 | in this reference Rinke K appears twice. Is it correct ? | |

I think that tables and Figures must be improved. The editing of lines in Table 1 and 2 must be definetly improved.
WHy do not add the DOC in table 1 ?

As far as Table 2 is concerned, this table is difficult to read due to an improper managing of spacing between lines. Please improve it.

As far as Figures are concerned, the Reference line in FIgure 1 is useless and could be deleted. Morover nowhere appears reference to Figure 1b

---

## Referee Comment (RC2) · Anonymous Referee #2 · 18 Mar 2016

This manuscript provides a relatively simple and straight forward approach to improve the estimation of water density from water temperature and electrical conductivity by including variables to incorporate the effects of different solutes in the waterbody. I think this material is of interest to the readers of HESS; however, I think that a few additional things could be added to significantly improve the manuscript prior to being officially published (described below).

Additional Discussion items that would be useful:

1. Given that the approach developed in this paper does require considerable water quality information, it would be useful to provide a suggestion on how to use the results of this paper to estimate the coefficients in other water bodies that do not have this detailed information.

2. The main benefit of this new approach appears to be an improvement in the absolute estimate of density. It would be helpful to discuss the absolute improvement in the density estimated versus the relative improvement. In other words, does this approach primarily shift the curves in Fig. 1 (first column) up and down? If this is the main improvement, it will not significantly change the results that have been obtained in most modeling exercises. I think that this discussion should be included.

General comments:

1. Most people refer to Mono as Mono Lake not Lake Mono.

2. In your comparison of methods, why is the most common approach the UNESCO approach not used for Rappbode, Geneva, and Constance. Even if it provides similar results to another mention, it should be at least mentioned.

Specific Comments:

Page 1. Modify the title to say "approach to estimating lake water density"

Page 1, line 14. Consider adding the word "absolute" in front of the word accuracy.

Page 1, line 21. Remove the words by far.

Page 2, line 6. Wouldn't it make sense to add seasonality as your main example?

Page 2, line 12. Add the word "do" between that and not.

Page 3. Line 5. I would delete this sentence

Page 3. Line 7. Why not include this sentence in the paragraph before this.

Page 3. Line 33. Consider adding (and lake specific variables describing the effects of differences in the chemical composition of the water) to the end of the sentence.

Page 4. Line 3. Change the word deliver to provide.

Table 1. Consider cutting back on the number of significant digits, unless they are real.

Table 2. There is no discussion of the starred lamda's in the table. If it is important it should be included in the paper. If not it should be deleted. Consider cutting back on the number of significant digits, unless they are real.

---

## Referee Comment (RC3) · Anonymous Referee #3 · 22 Mar 2016

*Review of*

**"A practical approach to lake water density
from electrical conductivity and temperature"**

*by S. Moreira, M. Schultze, K. Rahn and B. Boehrer*

In this paper a new method for computing the density of lake water is presented. It relies on finding the values of two constants, $\lambda_0$ and $\lambda_1$, after which the potential density can be estimated using measured conductivity. The authors claim an improved formula is required because other methods do not adequately take into account the variable composition of dissolved chemicals among different lakes. They show the resulting formula is better at predicting the density of water in a number of different lakes.

The paper is interesting and the new formula appears to be more accurate than other methods and is easy to apply. It does require either (i) measurements of the chemical composition of water in the lake or (ii) measurements of the density of water in the lakes at two different temperatures.

The paper could be strengthened by examples of when the improved density prediction matters. Who will benefit from the new formula? For example would numerical modelers of physical processes in lakes see any improvement using the new formula?

The writing could be improved in places. Some suggestions are listed below but there are many other places where the grammar could be improved a bit.

1. Abstract: Lines 12–13: "... and the conversion of measurements ...". Line 19: 'relative accuracy of 10%' should be 'relative error of less than 10%. Line 20: "which surmounts" should be "which is bettern than"

2. Lake Mono should be Mono Lake throughout the manuscript

3. Page 3, lines 31–33: The sentences "In conclusion ... conductivity" do not flow well with the preceding. Something more is needed to lead into these statements.

4. Page 9, lines 17–18: This sentence doesn't make sense to me. In the preceding you say that the equation is only applicable for temperatures up to 24°C. Why are you now taking about dissolved ions?

5. Page 9, Lines 24–25: Delete the last sentence. It repeats the factor of 2 mention in the first couple of sentences of this paragraph.

6. Page 10, lines 11-12: I don't understand the sentence "Large differences ....."'

7. Page 10, line 19: "at an accuracy of 10%" is not good - it says the results are not very good. Should say "with an error of less than 10%".

8. Page 10, lines 25: Do you mean "which can be measured in limnic waters"? What is meant by "delicate quantity"?

[revised manuscript text omitted]

---

## Author Response (AR1)

**Author's Response for the paper: "A practical approach to lake water density from electrical conductivity and temperature"**

by S. Moreira, M. Schultze, K. Rahn and B. Boehrer

**MS No.: hess-2016-36**

**Response to Reviewer1's comments (RC1 and SC1):**

*One might expect that, after calibration for a lake, simply measuring conductivity and temperature profiles could provide a density profile. As far as I understand, this is true only if the chemical composition does not change over the water column. Actually, eq. (2) depends on the composition of solutes...*

To be exact: Eq. 2 does not depend on composition; the value for lambda_0 depends on composition.

*… so that, if this quantity keeps changing over the depth, as it happens in weakly mixed deep lakes, it should be recomputed for each depth on the basis, e.g., of RHOMV. Can Authors better explain this point in the Discussion?*

Clearly, this numerical approach does not remove difficulties connected to chemical gradients in lakes, it only helps dealing with them. There are cases that pose big problems and some purposes will require accuracies that cannot be met with our approach. – We do not ignore this, but we claim for the majority of lakes, we provide an easily applicable density formula that improves calculation of solute density contributions by factors of typically 5 to 10 compared to usual approaches such as UNESCO (1983) or Chen & Millero (1986). Most lakes do NOT show pronounced gradients of chemical composition.

If there is the fear that the chemical conditions change too much within one lake for uniform lambdas (e.g. presented Waldsee), then the lambdas need to be evaluated for more than one water sample. In a second step, the various sets of lambdas can be compared with each other. For most purposes and most lakes, a uniform set of lambda coefficients will be accurate enough. Probably our paper is the first that presents a critical and quantitative assessment of the application of density formulas to more than a single lake.

To deal with the effect of local variability of solute composition, we did not use the

same sample for chemical evaluation and density measurement. In some cases, we used literature data for the chemical composition. Hence, the variability within one lake is included in our assessment. – The reviewer is probably right that we could emphasize this issue more in the discussion – especially, as a later comment shows that he did not grasp this argument in the discussion. – We included two sentences in the Discussion section: "Spatial and temporal variability of solute composition could contribute to errors in density calculation. However, where we attained chemical composition separately (i.e. from another sample) than the density information, this error is intrinsically included in our assessment and hence in the value that we supply for the RHO_LAMBDA approach."

*List of typos:*

All the corrections and suggestions mentioned in the list of typos have been dealt with as written here. There was just one comment we do not understand.

*Page 2, line 33:* It is a typo, the manual is Hodges and Dalimore (2007) accessed last time in 2014.

*Page 3, line 26:* $Si(OH)^{-4}$ has been corrected by $Si(OH)_4$

*Page 4, line 24:* the text "in the remaining part of this manuscript, eq(1), completed by eq(2) and eq(3), will be referenced as RHO_LAMBDA approach", has been to the manuscript added for clarity as suggested by the reviewer.

*Page 5, line 14:* the value has been corrected to 0.1635 mS cm$^{-1}$ rounding the value present in Table 2.

*Page 5, line 19: Here is $\rho_{MV}$ computed according to RHOMV?*

Yes, $\rho_{MV}$ is computed using RHOMV (Boehrer et al. 2010).

*Page 6, line 19: Probably in place of "our assessment section (Sect. 3)", "this section" would be better.*

We have changed to "this assessment section" and we have kept "(Sect. 3)" following the author's guidelines of the Journal.

*Page 6, line 21: This phrase could be improved, such as "an alternative approach to compute density, in order to check the accuracy of $\rho\lambda$".*

It has been changed to "a specifically obtained approach to density (e.g., Mono Lake or seawater) to check the accuracy of $\rho_\lambda$ in general", we hope it is more clear now.

*Page 7, line 3-5: "Hence..." Could you kindly explain better?*

The comment refers to this sentence: "On purpose, we obtained the chemical composition from a different source (sample) than the density measurement. Hence the error of variable water composition within one lake was included in our assessment." We changed the text to : "On purpose, we obtained the chemical composition from a different source (sample) than the density measurement. In this way, the variability of the water composition within one lake was included in the error determination in our assessment."

We hope that the content is clear: as above mentioned by Reviewer #1, local variabilities of solute composition will result in different lambda values, and hence to inaccurate density calculation. How big this effect is, is incorporated in the assessment, when density measurements and chemical composition are from different samples. In conclusion, the assessment of our lambda method is properly done including all sources of error such as local variability or inaccuracies of chemical analysis.

*Page 9, line 11:* "most of other of the approaches" corrected to "most of the other approaches"

*Page 15, line 24: in this reference Rinke K appears twice. Is it correct?*

Yes, there are two authors Rinke K, one is Karsten Rinke and the other one is Kristine Rinke.

*I think that tables and Figures must be improved. The editing of lines in Table 1 and 2 must be definitely improved.*

We improved the figures: axis are written in black now. Tables have been edited for appropriate line space.

*Why do not add the DOC in table 1 ?*

DOC is now included in table 1.

*As far as Table 2 is concerned, this table is difficult to read due to an improper managing of spacing between lines. Please improve it.*

We have corrected the formatting of Table 2 (specially the spacing between lines) to improve readability and included in the text the specific references to Figure 1b.

*As far as Figures are concerned, the Reference line in Figure 1 is useless and*

*could be deleted. Morover nowhere appears reference to Figure 1b.*

We have changed the axis labels to black in all figures. We hope this fulfils the requirements for improved Figure layout. The Reference line is a guide for the eye to give the reader a clear orientation of the zero line. Hence, we retain it. Figure 1b is referenced now.

**Response to Reviewer2's comments (RC2):**

*Additional Discussion items that would be useful:*

*1. Given that the approach developed in this paper does require considerable water quality information, it would be useful to provide a suggestion on how to use the results of this paper to estimate the coefficients in other water bodies that do not have this detailed information.*

This approach only requires the concentration of the major ions. This information is available in most limnologically studied water bodies. However, if not, the first option is taking a water sample and getting a reasonable idea of the water chemistry – This is not expensive. If no chemical data are included, density contributions of solutes are very badly represented. With a small effort, you can reduce the error by a factor 5 to 10. Even an analysis of limited accuracy will yield a much better density relation than UNESCO (1983) or Chen & Millero (1986).

*2. The main benefit of this new approach appears to be an improvement in the absolute estimate of density. It would be helpful to discuss the absolute improvement in the density estimated versus the relative improvement. In other words, does this approach primarily shift the curves in Fig. 1 (first column) up and down? If this is the main improvement, it will not significantly change the results that have been obtained in most modelling exercises. I think that this discussion should be included.*

We agree with the reviewer that it is important to represent density differences (gradients) accurately. Of course, we also improve the absolute value of density ("shifting up and down" in left column of Fig. 1). However, our approach is especially designed to represent density gradients due to solute gradients, which are so badly represented in the standard approaches.

In detail: Two water parcels of the same temperature (e.g. 15°C) but different solute concentration (0 and Lake Constance conc.) have different densities. Now looking at Fig. 1, right column, Lake Constance: the density difference between those water parcels is underestimated by about 6% using our lambda approach but by 45% using Chen & Millero (1986). The lambda approach is better by a factor of 8.

If solute concentrations differences are only a few percent of the difference of 0 to Lake Constance conc., this scales down roughly linearly. The relative inaccuracies remain roughly the same, and hence also the advantage of the lambda approach over Chen & Millero (1986).

In conclusion, yes, this lambda approach significantly improves the calculation of density stratification, if gradients of solute concentrations are involved.

*General comments:*

    *1. Most people refer to Mono as Mono Lake and not Lake Mono.*

    *2. In your comparison of methods, why is the most common approach the UNESCO approach not used for Rappbode, Geneva, and Constance. Even if it provides similar results to another mention, it should be at least mentioned.*

    1. We accept the correction of the reviewer and we will modify all the references the text from "Lake Mono" to "Mono Lake".

    2. UNESCO (1983) is only valid above a salinity of 2 psu. For the case below 2psu, Chen & Millero (1986) replaced the UNESCO formula by their slightly different approach, trying to remove some short comings of ocean salinity at very low values. – do not expect any better results from the Unesco formula. We are aware that most numerical models use UNESCO for freshwater, despite the fact that it is not recommended. However, using UNESCO in this critical comparison would mean using and blaming the formula for conditions it is not made for. We wanted to avoid this.

*Specific Comments:*

*Page 1: Modify the title to say: "approach to estimating lake water density"*

This is not guessing. We present an approach for accurate calculation of density in limnic waters and we even provide the detailed assessment of its accuracy. We retain our title.

*Page 1, line 14: consider adding "absolute" in front of the word "accuracy".*

Considered but we do not see the implication of the word. What feature is the absoluteness of accuracy? Hence, not included.

*Page 1, line 21: remove the words "by far".*

Done.

*Page 2, line 6: wouldn't it make sense to add seasonality as your main example?*

Sure, one can investigate this, but a seasonality of the lambda coefficients can only be expected for extreme cases, where the composition of the solutes changes dramatically. We do not have such a lake in our focus.

*Page 2, line 12: Add the word "do" between that and not.*

Done.

*Page 3, line 5: I would delete this sentence.*

We eliminated those lines and rephrased the paragraph.

*Page 3, line 7: Why not include this sentence in the paragraph before this?*

Done.

*Page 3, line 33: Considering adding (and lake specific variables describing the effects of differences in the chemical composition of the water) to the end of the sentence.*

Sorry, we do not understand.

*Page 4, line 3: change the word "deliver" to "provide".*

Done.

*Table 1, consider cutting back on the number of significant digits, unless they are real.*

We have reduced the number of significant digits in the calculated conductivity, but we will keep them in the measured variables of the table. Three decimal digits have been kept in all density values. Due to the small differences between the Practical Salinity, Absolute Salinity and Corrected Salinity, the decimals have been kept too.

*Table 2. There is no discussion of the starred lambdas in the table. If it is important it should be included in the paper, if not it should be deleted. Consider cutting back on the number of significant digits, unless they are real.*

In the caption of Table 2, on can read "... , while $\lambda_0^*$ and $\lambda_1^*$ represent the lambda values obtained from a linear regression of the density reference"
A small paragraph about starred lambdas has been added: "The values of $\lambda_0$ and $\lambda_1$ have also been calculated using direct measurements of density (starred values, $\lambda_0^*$ and $\lambda_1^*$). In the case of $\lambda_0$ only slight differences can be found between the values calculated from chemical composition and from direct measurements of density. However, those differences increase in the case of $\lambda_1$, as mentioned above."

Accepted; we show two significant digits of all lambdas. The starred values are the empirical values from density measurements. Both, starred and non-starred values are shown in figure 2 to demonstrate the accuracy quantitatively.

**Response to Reviewer3's comments (RC3):**

*The paper could be strengthened by examples of when the improved density prediction matters. Who will benefit from the new formula? For example would numerical modellers of physical processes in lakes see any improvement using the new formula?*

EVERYBODY, who uses the more accurate approach will profit from more accurate results. In all presented lakes (not the ocean) the density contribution of solutes is out by 25% to 60% according to UNESCO (1983) or Chen & Millero (1986). Hence, ALL investigations of stratification features due to solute gradients will profit from the better approach. We demonstrated that for all lakes in this manuscript; it will be very difficult to find a lake where the situation lies outside our extremely wide band of considered lakes (if the lake water is not dominated by ocean water). On top of it, the lambda approach is much easier to implement.

We admit that the implementation of this better knowledge into numerical models is not provided. This is outside the scope of this paper. Numerical models use salinity to quantify solute concentration, our lambda approach avoids salinity because it is a precarious quantity in limnic waters. We base density directly on electrical conductivity. However, using electrical conductivity in numerical models is not straight forward (It does not mix linearly at high concentrations).

*The writing could be improved in places. Some suggestions are listed below but there are many other places where the grammar could be improved a bit.*

> *1. Abstract: Lines 12–13: "... and the conversion of measurements ...". Line 19: 'relative accuracy of 10%' should be 'relative error of less than 10%. Line 20: "which surmounts" should be "which is better than" .*

Done. Now it reads: "The results show that our new approach can reproduce the density contribution of solutes with a relative error of less than 10% in lake waters from very low to very high concentrations as well as in lakes of very particular water chemistry, which is better than all commonly implemented density calculations in lakes."

> *2. Lake Mono should be Mono Lake throughout the manuscript*

We accept the correction of the reviewer and we will modify all the references in the text from "Lake Mono" to "Mono Lake".

3. *Page 3, lines 31–33: The sentences "In conclusion ... conductivity" do not flow well with the preceding. Something more is needed to lead into these statements.*

We have moved it to the next paragraph.

4. *Page 9, lines 17–18: This sentence doesn't make sense to me. In the preceding you say that the equation is only applicable for temperatures up to 24 ◦ C. Why are you now taking about dissolved ions?*

We wanted to point out the fact that the error using Bührer and Ambühl (1975) in temperatures higher than 24°C becomes smaller and smaller when the solute concentration increases. We have rephrased the text for clarity. We have rephrased the text for clarity. It reads now: "The observed strong increase of the relative error with temperature for Bührer and Ambühl (1975) was caused by its validity limited to 24°C."

5. *Page 9, Lines 24–25: Delete the last sentence. It repeats the factor of 2 mention in the first couple of sentences of this paragraph.*

Done.

6. *Page 10, lines 11-12: I don't understand the sentence "Large differences .....'*

Further explanation has been added to the text. "However, the difference posed the question of how accurately the shift of temperature of maximum density would actually be indicated by coefficients of physical chemistry literature. The largest discrepancies appeared for freshwater lakes (Rappbode Reservoir, Lake Geneva, Lake Constance) where the shift is small."

7. *Page 10, line 19: "at an accuracy of 10%" is not good - it says the results are not very good. Should say "with an error of less than 10%".*

Done.

8. *Page 10, lines 25: Do you mean "which can be measured in limnic waters"? What is meant by "delicate quantity"?*

We have replaced "verified" by "measured" in the text.
We changed to: "which is badly defined for limnic waters and hence a precarious quantity".